# A Narrative Review of Lifestyle Risk Factors and the Role of Oxidative Stress in Age-Related Hearing Loss

**DOI:** 10.3390/antiox12040878

**Published:** 2023-04-04

**Authors:** Diana Tang, Yvonne Tran, Piers Dawes, Bamini Gopinath

**Affiliations:** 1Macquarie University Hearing, Faculty of Medicine Health and Human Sciences, Macquarie University, Sydney, NSW 2109, Australia; yvonne.tran@mq.edu.au (Y.T.);; 2Centre for Hearing Research, School of Health and Rehabilitation Sciences, University of Queensland, St. Lucia, QLD 4072, Australia

**Keywords:** hearing loss, adults, oxidative stress, lifestyle, noise, ototoxicity

## Abstract

Age-related hearing loss affects a significant proportion of adults aged 60 years and above, with a prevalence of 65%. This condition has a negative impact on both physical and mental well-being, and while hearing interventions can help alleviate the effects of hearing loss, they cannot completely restore normal hearing or halt the progression of age-related hearing loss. Oxidative stress and inflammation have been identified as potential contributors to this condition. By addressing modifiable lifestyle risk factors that exacerbate oxidative stress, there may be an opportunity to prevent hearing loss. Therefore, this narrative review provides an overview of the major modifiable lifestyle risk factors associated with age-related hearing loss, that is, exposure to noise and ototoxic chemicals, smoking, diet, physical activity, and the presence of chronic lifestyle diseases, and offers an overview of the role of oxidative stress in the pathophysiology of this condition.

## 1. Introduction

Hearing loss is a global health issue that affects approximately one in five individuals, making it the third most significant cause of years lived with disability worldwide [1,2]. Although regarded as an ‘invisible disability’ due to a lack of visible symptoms, public stigma, and disregard by policymakers [1,3], the condition is both prevalent and debilitating. The negative impacts on communication lead to poorer quality of life and psychosocial health [1,4,5,6], depression [7], and increased mortality [8], with hearing loss also frequently associated with tinnitus, cognitive decline, and dementia [9,10]. A prevalent type of hearing loss is age-related hearing loss (ARHL), also known as presbycusis, affecting 65% of adults aged ≥60 years, with 25% experiencing moderate or worse hearing loss (≥35 dB HL in the better hearing ear). It is characterised by bilateral and symmetrical loss of high frequency hearing (≥8 kHz) in the initial stages, with progression of the condition affecting hearing at lower frequencies due to the irreversible loss of cochlear hair cells and damage to the auditory nerves that are crucial for hearing [11]. The development of ARHL is influenced by a combination of aging, health comorbidities, lifestyle, environmental, and genetic factors [12,13,14,15]. These factors promote inflammation-related processes that produce reactive oxygen species (ROS) [16]. When there is an overproduction of ROS and insufficient endogenous processes to neutralize or detoxify them, it results in a disruption of intracellular redox homeostasis, leading to oxidative stress. This oxidative stress is believed to cause damage to delicate inner ear structures via peroxidation of membrane lipids and protein denaturation, as well as DNA damage and cellular apoptosis [14,15,16].

Given the increasing prevalence of ARHL due to ageing demographics, there is a need for public health strategies aimed at reducing oxidative damage to manage the severity and burden of this condition. Although some factors contributing to oxidative stress are non-modifiable, there are several modifiable lifestyle risk factors that can be managed to promote antioxidant activity. These lifestyle risk factors include exposure to noise and ototoxic chemicals, smoking, a lack of regular physical activity, a poor diet, and the presence of chronic lifestyle diseases. This review will provide a comprehensive review of the role of these modifiable lifestyle risk factors in the development and progression of ARHL, including their proposed mechanism of increasing oxidative stress. Many of the studies included in this review report on risk ratios. This includes relative risk (RR), which compares the risk of one group relative to another, odds ratio (OR), which indicates the strength of association between one variable relative to another, and hazard ratio (HR), which compares the rate of change between one group relative to another [17]. Additionally, recommendations to manage these risk factors to potentially reduce the incidence and severity of ARHL will be discussed.

### Search Strategy and Selection Criteria

To gather information for this review, a comprehensive literature search was performed in Medline and Embase via the OVID platform, in addition to Google Scholar. The search strategy involved using the keywords “hearing loss” or “presbycusis” in combination with “oxidative stress” and “lifestyle”, “noise”, “ototoxic”, “smoking”, “diet”, or “exercise”, and “risk”. The search was not limited by publication year. Relevant full-text articles written in English were collected and reviewed. Additionally, reference lists of relevant articles were examined to identify any additional suitable articles for inclusion in the review.

## 2. Mechanisms of Oxidative Stress

Ageing is the biggest risk factor for ARHL and is characterised by a state of chronic oxidative stress and inflammation [1,18]. Broadly, lifestyle risk factors such as exposure to noise and ototoxic chemicals, smoking, a lack of physical activity, a poor diet, and the presence of chronic lifestyle diseases can create stressful environments that exacerbate the pro-oxidant and pro-inflammatory processes in the body (Figure 1). These processes are interrelated and are therefore jointly present in a number of chronic diseases, including ARHL [19]. When pro-oxidant activity exceeds antioxidant activity, toxic levels of ROS and other free radicals are produced. This increases the likelihood of damaging interactions with macromolecules, including lipids, proteins, and DNA within the auditory cells (e.g., hair cells, spiral ganglion neurons) [14,20,21]. These interactions are particularly damaging to the cells’ mitochondria and mitochondrial DNA as ROS inflicts peroxidative damage to the mitochondrial membrane and reduces the mitochondria’s cytochrome C oxidase activity [15,18,22]. These reactions lead to a spiral of oxidative damage as dysfunctional mitochondria then produce more ROS, triggering premature metabolic senescence [14,22] and apoptosis [15], leading to the progressive and permanent loss of auditory cells [15,22].

Oxidative stress and inflammation can also cause damage to the endothelium [15]. Maintaining the health of the endothelium is important for successful ageing as it is responsible for releasing enzymes and/or nitric oxide to maintain vascular tone, control platelet adhesion, blood clotting, and vascular proliferation [15,21]. Oxidative damage to the endothelial cells of the stria vascularis due to poor vascular health, has also been linked to the pathophysiology of ARHL [14]. The stria vascularis is a structure within the cochlea that has a critical role in hearing and cochlear amplification [23]. In addition to ageing and poor vascular health, oxidative stress can be induced by the presence of other health comorbidities (e.g., obesity, diabetes), environmental (e.g., noise and ototoxic chemical exposure), and genetic factors [15].

## 3. Modifiable Lifestyle Risk Factors and Oxidative Stress

ARHL is influenced by various lifestyle factors that can be modified to reduce the risk and progression of the condition. These modifiable factors include exposure to noise and ototoxic chemicals, smoking, an unhealthy diet, a lack of regular exercise, and the presence of chronic lifestyle diseases such as obesity, diabetes, and other cardiovascular health issues [13,18,24,25,26,27,28,29,30,31]. Research literature suggests that the likely mechanism of action is related to oxidative damage and inflammation [14,15,22] where healthy lifestyle behaviours such as safe listening, smoking cessation, a healthy diet, and regular physical activity have been shown to reduce the risk and progression of ARHL [18,26,27,28,29,30,31,32,33,34,35,36,37,38,39,40,41,42]. In this section of the review, we will present a comprehensive summary of the major modifiable risk factors associated with ARHL, their proposed mechanisms of increasing oxidative stress, and recommended strategies to manage and mitigate these risk factors. Table 1 provides an overview of the risk factors discussed and the key sources of research evidence demonstrating the association.

### 3.1. Noise Exposure

According to the World Health Organisation (WHO), exposure to noise louder than 80 dB for more than 40 h per week can result in noise-induced hearing loss, which may progress to ARHL [1]. Such noises can be heard in occupational, recreational, and environmental settings, including building and construction, mining, concerts, personal music devices, traffic, and home appliances [1]. The damage caused by noise can occur at both mechanical and metabolic levels in the inner ear [16]. The mechanical damage can include the loss of stereocilia bundles in auditory hair cells and the disruption of cellular membranes and organelles due to acoustic overstimulation and high-intensity noise. This damage triggers significant metabolic activity that can lead to ROS formation, reduce blood flow to the cochlea, and result in cellular apoptosis [16,63].

Research evidence generally supports the association between noise exposure and hearing loss. Data from two cross-sectional population-based cohorts (*n* = 27,580 and *n* = 26,606, respectively) from the Trøndelag Health (HUNT) Study, investigated two decades apart, found that exposure to recreational firearms was associated with worse hearing at 3–6 kHz, with a stronger association observed with increasing number of lifetime shots and age [43]. The Blue Mountains Hearing Study showed that occupational noise exposure for >10 years in adults aged ≥50 years, significantly increased the odds of ARHL by more than 2-fold (OR 2.39, 95% CI 1.37–4.19), and of moderate-to-severe ARHL by almost 7-fold (OR 6.80, 95% CI 2.97–15.60) [44]. In the Framingham Heart Study, noise-damaged ears showed greater hearing loss decline than non-damaged ears after 15 years [45]. The authors, Gate et al., suggested that noise-related damage may persist long after the initial exposure [45]. However, other longitudinal studies such as the Epidemiology of Hearing Loss Study [46] and Hederstierna et al.’s study [47] of 1013 Swedish older adults were not able to confirm this association.

To prevent the risk of hearing damage from noise exposure, the primary recommendation is to minimise exposure to loud sounds and noises [1]. A strategy to achieve this includes participating in hearing preservation initiatives which provide education and promote safe listening [1]. Self-monitoring noise exposure levels through services such as the ‘Know Your Noise’ website can also provide personalised advice regarding hearing health [64]. Additionally, the use of hearing protection such as headphones and earplugs are recommended [1]. For occupations that frequently expose individuals to noise, regular hearing assessments to monitor staff hearing health should also be considered [65].

### 3.2. Ototoxic Chemical Exposure

ARHL can be exacerbated by ototoxic chemicals such as pharmaceutical drugs, solvents, asphyxiants, and heavy metals through oxidative stress-inducing pathways [65,66]. These chemicals can cause damage to the auditory system, including hair cells, spiral ganglion cells, and auditory nerves, as they enter the bloodstream and reach the cochlea or central nervous system [65,66]. The mechanism of action of ototoxic chemicals has been linked to the formation of ROS and oxidative damage, with hair cells being the most susceptible to ototoxicity [63,65,66]. For example, aminoglycoside antibiotics interact with polyphosphoinositides to increase the permeability of cell membranes, triggering a process that releases ROS [65]. Cisplatin, an anticancer agent, stimulates the activity of the reduced nicotinamide adenine dinucleotide phosphate (NADPH) oxidase 3 (NOX3) enzyme, which has a role in ROS production [66]. Solvents and asphyxiants are oxidising chemical agents that cause oxidative damage and hair cell loss [63], while prolonged exposure to lead induces oxidative stress in the cochlea by downregulating the production of antioxidant enzymes and upregulating processes involved in cellular apoptosis [67].

Co-exposure to ototoxic chemicals and noise also increases hearing loss risk in older adults [48,49]. Among 547 adults aged 65–75 years, lifelong exposure to ototoxic chemicals and/or noise significantly increased the likelihood of hearing loss by 2-fold (OR = 2.29; 95% CI: 1.17 = 4.51) when compared to non-exposed adults [48]. Exposure to ototoxic solvents has also been specifically reported to increase susceptibility to noise-induced hearing loss [49,65]. This is supported by a systematic review and meta-analysis of 13 studies, which showed that the pooled odds of hearing loss among adults in the workplace were more than 2-fold higher with combined exposure to ototoxic solvents and noise than to solvent exposure alone (pooled OR = 2.15, 95% CI: 1.24–3.72, *p* = 0.006) [49].

Similar to managing noise exposure, the primary recommendation for addressing ototoxicity is to avoid exposure to ototoxic chemicals. While it may be difficult to avoid exposure to ototoxic drugs, as they are often used to treat life-threatening conditions, multidisciplinary care can help reduce co-exposures to noise that can exacerbate hearing loss [65]. In the workplace, hazard assessment and control protocols can reduce exposure risks to other ototoxic chemicals. This may involve removing the source of risk, implementing strategies to minimise exposure, and/or enforcing the use of personal protective equipment [65]. As with occupations involving regular noise exposure, those involving ototoxic chemical exposure should also consider providing regular hearing assessments to monitor the hearing health of staff [65].

### 3.3. Smoking

Smoking directly increases oxidative stress and triggers an inflammatory response in the body [30,65]. This is because cigarette smoke contains ototoxic chemicals such as hydrogen cyanide and ROS, as well as water-soluble components that can circulate throughout the body, causing systemic oxidative stress [30]. As a result, smoking has been shown to increase the risk of hearing loss in a number of studies. In the Blue Mountains Study (Australia), involving 2815 adults aged ≥50 years, those who were current smokers were 61% (OR 1.61; 95% CI = 1.02 to 2.53) more likely to have existing (prevalent) hearing loss after multivariate adjustment accounting for confounding factors such as noise exposure [39]. The Epidemiology of Hearing Loss Study (USA), involving 1678 adults aged 43–84 years, also showed that smoking significantly increased the risk of developing (incident) hearing loss by 31% (HR = 1.36, 95% CI = 1.05, 1.77), after 15 years [33]. A higher risk of hearing loss among female current smokers versus female never smokers was also reported in the Conservation of Hearing Study, as over 1,533,214 person-years of smokers had a 21% (RR = 1.21, 95% CI = 1.02, 1.43) increased risk. The increased risk was higher with a greater number of pack-years smoked (RR = 1.30, 95% CI = 1.09, 1.55) [34]. In addition, there is also evidence showing that passive smokers are at increased risk of ARHL [32]. Data from 164,770 adults aged 40–69 years from the UK Biobank Resource demonstrated a dose-dependent response as longer periods of passive smoke exposure consistently resulted in higher odds of ARHL: ≤1 h/week of passive smoke exposure showed no additional risk (OR = 1.00, 95% CI = 0.94, 1.07), 2–9 h/week showed a 28% increased risk (OR = 1.28, 95 % CI = 1.18, 1.39), and ≥ 10 h/week showed a 39% increased risk (OR = 1.39, 95% CI = 1.19, 1.61) [32].

On the other hand, smoking cessation has been shown to increase the concentration of protective antioxidants in the plasma and improve resistance to oxidative challenge [31]. This may explain the observed risk reduction reported from the Epidemiology of Hearing Loss Study where former smokers who quit for <5 years had slight but non-significant increased risk (HR = 1.24, 95% CI = 0.88, 1.75), while those who quit for >5 years had reduced their risk close to a never smoker’s level (HR = 1.10, 95% CI = 0.91, 1.35) [33]. These benefits were also reported in the Conservation of Hearing Study, where increasing years of smoking cessation were associated with greater risk reduction: smoking cessation for <5 years: RR = 1.43 (95% CI = 1.17, 1.75), smoking cessation for 5–9 years: RR = 1.27 (95% CI = 1.03, 1.56), and smoking cessation for 10–14 years: RR = 1.17 (95% CI = 0.96, 1.41) [34].

### 3.4. Dietary Factors

#### 3.4.1. Diet Quality and Eating Patterns

Overall diet quality is associated with hearing loss [35]. The National Health and Nutrition Examination Study (NHANES) evaluated diet quality using the Healthy Eating Index (HEI), which accounts for intake of the five core food groups (grains, fruits, vegetables, meat, and dairy), four nutrients (fat, saturated fat, cholesterol, and sodium), and overall food variety [35]. Cross-sectional NHANES data from 2366 adults aged 20–69 years showed that better diet quality (higher Healthy Eating Index scores) was associated with lower (better) high-frequency hearing thresholds (Wald F = 6.54, df = 429; *p* ≤ 0.05) [35]. A high-quality diet such as the Mediterranean diet is a rich source of antioxidants and micronutrients that can inhibit inflammation and reduce oxidative stress [68]. This diet consists of nutrient-rich vegetables, fruit, fish, olive oil, and moderate wine intake [68]. A recent cross-sectional study involving the completion of laboratory tests and questionnaires by 335 older adults showed that participants with good diet quality had a high total antioxidant capacity [69], which supports the role of oxidative stress and inflammation on the development and progression of ARHL. Contrastingly, an opposing eating pattern that is energy-dense, high in saturated fat, salt, and sugar, and low in fibre, increases oxidative stress and the risk of obesity, type II diabetes, and cardiovascular disease [68].

#### 3.4.2. Nutrients

Research evidence in adults supports associations between specific nutrients, ARHL, and oxidative and inflammatory pathways [27,28,29,53]. For example, vitamins A, C, and E are suggested to reduce oxidative stress by acting as free radical scavengers in synergy with magnesium to counter pro-oxidant activity and maintain healthy concentrations of nitric oxide [27]. Longitudinal data from the Blue Mountains Hearing Study concluded that vitamins A and E were significantly associated with the prevalence but not incidence of ARHL [50]. That is, the highest quintile of vitamin A intake compared to the lowest quintile of intake was associated with a 47% (OR = 0.53, 95% CI = 0.30, 0.92) reduction in the risk of having moderate or worse hearing loss (>40 dB HL), and each standard deviation increase in vitamin E intake was associated with a 14% (OR = 0.86, 95% CI = 0.78, 0.98) reduction in prevalent hearing loss [50]. Higher intakes of vitamin A, in the form of retinol, were also found to be significantly associated with lower (better) hearing thresholds in a cohort of French women based on a mean difference of −1.79 decibels hearing level (HL) (95% CI = −3.26, −0.32) compared to the lowest quartile of retinal intake (reference) [51]. The findings in French men were not significant [51]. In addition, this study observed improvements in hearing thresholds with higher intakes of vitamin B12 compared to lower intakes (mean difference: −1.57 HL, 95% CI = −3.04, −0.09) [51]. The authors did not observe associations with other micronutrients such as beta-carotene, vitamin C, E, B6, and B9 (folate) [51]. Contrastingly, NHANES data found that higher intakes of beta-carotene, vitamin C, and magnesium were independently associated with lower (better) PTAs based on percentage changes of hearing thresholds at speech frequencies (beta-carotene, −14.31% (95% CI = −21.03, −7.03); vitamin C, −14.17 (95% CI = −22.04, −5.50); magnesium, −13.82% (95% CI = −21.07, −5.90)); and high frequencies (beta-carotene, −14.46% (95% CI = −20.83, −7.59); vitamin C, −12.31 (95% CI = −19.69, −4.26); magnesium, −11.93% (95% CI = −20.02, −3.02). Only the highest quartile of vitamin E intake was associated with lower PTAs [52]. Shargorodsky et al.’s study involving 3559 male cases of hearing loss, aged 40–74 years, was not able to confirm the benefits of vitamin A, C, and E for hearing loss, but instead identified that higher intakes of dietary vitamin B9 (folate) reduced the risk of hearing loss development by 21% (HR = 0.79, 95% CI = 0.65, 0.96) in men 60 years and over [53].

Few studies have investigated the links between dietary flavonoids and ARHL in adults. Blue Mountains Hearing Study data found that higher intakes of one flavonoid subclass, isoflavone, were associated with a significant reduction in the risk of developing ARHL by 36% (OR = 0.64, 95% CI = 0.42, 0.99) [54]. However, this is likely a chance finding, as this flavonoid subclass is predominantly sourced from soy products, which were not well captured in the study, and associations with other subclasses were not observed [54]. Other research has reported that high plasma levels of omega-3 fatty acids were significantly associated with less hearing loss at low frequencies (2–4 kHz; mean difference hearing thresholds: −1.2 dB) after 3 years in a group of 720 older adults in The Netherlands [55]. The research evidence is less clear regarding the benefits of omega-3 supplement for ARHL [37,70]; however, regular fish consumption (2–4 times/week) versus consumption of less than once/month has been shown to be protective against hearing loss in women (RR = 0.80, 95% CI = 0.74, 0.88) in addition to higher intakes (5th quintile) of polyunsaturated fatty acids compared to lower intakes (1st quintile) (RR = 0.85, 95% CI = 0.80, 0.91) [37]. Fish are a natural source of omega-3 fatty acids, which can enhance the production of antioxidant nitric oxide, reduce oxidative DNA damage, and reduce the concentration of inflammatory cytokines [68].

Carbohydrate nutrition has also been investigated in relation to the prevalence and incidence of ARHL [36]. When comparing the highest versus lowest quintiles of intake, the greatest likelihood of prevalent ARHL was a high mean dietary glycaemic index (OR = 1.41, 95% CI = 1.01, 1.97) 36. Although this finding did not persist for the overall study population after adjusting for cereal fibre, the likelihood of hearing loss increased in those aged ≥70 years (OR = 1.67, 95% CI = 1.04, 2.69) [36]. Joint analysis of glycaemic index and cereal fibre intake also showed that adults with the least healthy diets across these categories (i.e., highest mean GI and lowest cereal fibre intake) were 70% (OR = 1.70, 95% CI = 1.07, 2.69) more likely to have mild ARHL (26–40 dB HL) 36. For incident hearing loss over 5 years, risk increased by 76% (OR = 1.76, 95% CI = 1.03, 3.00) and 23% (OR = 1.77, 95% CI = 1.04, 3.00) with the highest quartile of mean glycaemic load and carbohydrate intake, respectively [36]. Among adults aged <70 years but not older, a risk reduction of at least 62% was observed with lower versus higher sugar and carbohydrate intakes (OR = 0.28, 95% CI = 0.11, 0.70, and OR = 0.38, 95% CI = 0.17, 0.87, respectively [36]. These associations with ARHL may reflect a mechanistic link involving oxidative damage. Strong positive correlations between oxidative damage and sugar intake have been observed in other studies [71,72]. This is due to the processes of glycation and glycoxidation of sugars through the Maillard reaction, which result in the production of ROS [73].

#### 3.4.3. Beverages

Alcohol is associated with alcohol-induced oxidative stress through the process of ethanol metabolism, which produces ROS [74]. However, moderate consumption of alcoholic beverages is proposed to benefit vascular health [75]. Red wine is rich in antioxidants, particularly resveratrol, which exhibits a number of properties to reduce oxidative stress [75]. This includes antioxidant activity to scavenge free radicals and prevent the oxidation of low-density lipoprotein (LDL) cholesterol, anti-inflammatory activity by inhibiting the production of pro-inflammatory molecules and promoting nitric oxide production [75]. Through these processes, moderate alcohol consumption has been linked to a reduced risk of hearing loss [32,38,39,40]. Evidence from the UK Biobank study showed comparable risk reductions of approximately 40% for all levels of consumption [32], while Blue Mountains Hearing Study data observed reduced risk of any hearing loss prevalence and severe hearing loss prevalence with ≤1 standard drink (OR = 0.75, 95% CI = 0.57, 0.98), and >2 standard drinks (OR = 0.54, 95% CI = 0.31, 0.94), respectively. However, these associations did not persist longitudinally [39]. In Fransen et al.’s multicentre study, a protective association was observed in all included studies, although only three out of nine reached significance [38].

The UK Biobank study also provided some longitudinal evidence that coffee consumption in men but not women is protective against hearing loss. Among 343 men, those who consumed 1 or ≥2 cups/day equally lowered their risk of hearing loss by 28% (HR = 0.72, 95% CI = 0.54, 0.95; HR = 0.72, 95% CI = 0.56, 0.92, respectively) [40]. As coffee contains a number of antioxidant bioactives, a protective association against ARHL is plausible. In a review of intervention studies, Martini et al.’s findings suggested that regular coffee intake increases glutathione levels and protects against DNA damage [76].

### 3.5. Exercise

Compared to other modifiable lifestyle risk factors, fewer studies have explored the relationship between exercise and ARHL risk; however, the findings are promising. For example, among women in the Nurses’ Health Study II, higher physical activity intensity (4th and 5th quintile of metabolic-equivalent tasks (METs) in h/week) significantly reduced hearing loss risk by 14% (RR = 0.86, 95% CI = 0.81–0.91) and 17% (RR = 0.83, 95% CI = 0.78–0.88), respectively. Walking at least 2 h/week also significantly reduced hearing loss risk by 10% (RR = 0.90, 95% CI = 0.85–0.95), with an additional 5% risk reduction among women who walked ≥4 h/week. Similar benefits were observed in the Niigata Wellness Study in Japan with muscular and performance fitness in 21,907 adults aged 20–79 years [42], and leisure-time physical activity in 27,537 adults within the same age range [41]. A significant inverse dose-dependent response was observed between increasing muscular and performance fitness levels (from quartile 1 (lowest) or 4 (highest)) and risk of hearing loss; quartile 1 (reference): RR = 1.00; quartile 2: RR = 0.88 (95% CI = 0.79, 0.97), quartile 3: RR = 0.83 (95% CI = 0.75, 0.93), and quartile 4: RR = 0.79 (95% CI = 0.71, 0.88) [42]. Risk of hearing loss is also significantly reduced with the highest level of intensity (≥525 MET-min/week; HR = 0.87, 95% CI = 0.81, 0.95) and duration of physical activity (≥120 min/week; HR = 0.89, 95% CI = 0.82, 0.96) [41].

The benefits of regular exercise on ARHL are likely due to reductions in oxidative stress and inflammation associated with ageing through the management of obesity, sarcopenia, and mitochondrial dysfunction [18,77]. Specifically, exercise reduces the amount of visceral fat, which is responsible for releasing proinflammatory cytokines and stimulating the release of adiponectin, a substance that has anti-inflammatory, anti-oxidative, and anti-apoptotic properties [18]. Previous research has reported that low levels of adiponectin are associated with high-frequency ARHL in adults aged 40–86 years [78]. Exercise also reduces sarcopenia-induced inflammation and pro-oxidant activity by increasing muscle mass and strength to reduce injury risk—the trigger for releasing these substances [18]. There is also evidence suggesting that aerobic exercise in younger and older people supports the processes involved in mitochondrial biogenesis [77]. This includes promoting the acute release of stress signals, which activate pathways to transcribe the genes needed for biogenesis [77]. These findings highlight the importance of exercise in hearing interventions, as oxidative damage to the mitochondria is proposed to initiate the cascade of processes contributing to the progressive loss of auditory cells [22].

### 3.6. Lifestyle Diseases

Engaging in unhealthy lifestyle behaviours has been linked to the development of lifestyle diseases, such as obesity, diabetes, and cardiovascular disease. Studies have shown that the presence of these conditions may heighten the risk of ARHL [24,25,59,60,61,62]. This section will provide an overview of the research evidence on these lifestyle diseases with ARHL.

#### 3.6.1. Obesity

Yang et al.’s meta-analysis included 14 studies (*n* = 489,354 participants and 55,410 cases) [24]. The cross-sectional studies in this review supported a significantly increased likelihood of having a hearing loss among obese adults (OR = 1.40, 95% CI = 1.14, 1.72), with each 5 kg/m^2^ increase in BMI (OR = 1.14, 95% CI = 1.04, 1.24), and a larger waist circumference (OR = 1.22, 95% CI = 0.88, 1.68) [24]. Another cross-sectional study also showed that visceral fat increased the risk of hearing loss only in Korean women but not in men [79]. Indicators of obesity were also significantly associated with an increased risk of developing hearing loss [24]. Meta-analyses of longitudinal studies showed that being overweight or obese increased the risk of developing a hearing loss by 15% (RR = 1.15, 95% CI = 1.04, 1.27) and 38% (RR = 1.38, 95% CI = 1.07, 1.79), respectively, and each 5 kg/m^2^ increase in BMI and larger waist circumference were associated with a 15% (RR = 1.15, 95% CI = 1.01, 1.30) and 11% (RR = 1.11, 95% CI = 1.01, 1.22) increase in hearing loss risk, respectively. One of the included studies was the Nurses’ Health Study II, which showed a dose-dependent association between BMI and self-reported hearing loss in adults within obese class I (BMI 30–34 kg/m^2^) had a 17% (RR = 1.17, 95% CI = 1.10, 1.24) increased risk compared to 22% (RR = 1.22, 95% CI = 1.12, 1.31) for obese class II (BMI 35–39 kg/m^2^) and 25% (RR = 1.25, 95% CI = 1.14, 1.37) for obese class III (BMI ≥40 kg/m^2^) [56]. Similarly, hearing loss risk significantly increased from 11% (RR = 1.11, 95% CI = 1.02, 1.21) for a waist circumference of 80–88 cm to 27% (RR = 1.17, 95% CI = 1.17, 1.38) for a waist circumference of >88 cm [56].

There were also mixed findings from two studies investigating underweight and hearing loss risk; one study observed a 22% increased risk (RR = 1.22, 95% CI = 1.13, 1.31) [57], while the other reported non-significant protective effects (RR = 0.63, 95% CI = 0.52, 1.79) [58].

#### 3.6.2. Diabetes

Evidence from Horikawa et al.’s meta-analysis of 13 cross-sectional studies (*n* = 20,194 participants and 7377 cases) showed that the overall pooled odds of having any hearing loss was 2.15 (95% CI = 1.72–2.68) times greater among diabetic adults than non-diabetics [59]. The higher odds persisted following various subgroup stratifications, e.g., the odds of unilateral (OR = 2.21, 95% CI = 1.55–3.15) and bilateral hearing loss (OR = 2.10 (95% CI = 1.54–2.87) was more than two-fold in diabetics versus non-diabetics, and 2.61 times (OR = 2.61, 95% CI = 2.00–3.4) higher among younger diabetic adults (mean age ≤60 years) than older diabetic adults (mean age >60 years) compared to non-diabetic counterparts [59].

A second meta-analysis specifically focused on type 2 diabetes and incident hearing loss was conducted by Akinpelu et al. [60]. This review included 18 studies of prospective cohort, cross-sectional, or case-controlled design. The odds of developing hearing loss were almost two-fold among type 2 diabetics versus non-diabetic controls (OR = 1.91, 95% CI = 1.47, 2.49) [60]. Odds were also higher among younger diabetic adults (OR = 2.10, 95% CI = 1.22, and 3.64), than older diabetic adults (OR = 1.75, 95% CI = 1.46, 2.09), versus their non-diabetic counterparts [60]. A comparison of mean PTA thresholds across 0.5, 1, 2, 4, 6 and 8 kHz showed that thresholds were consistently higher (worse) among the diabetic group than the non-diabetic group, and both groups’ PTAs generally increased (worsened) with higher frequencies, up to 60 dB HL [60].

#### 3.6.3. Cardiovascular Disease

Research literature suggests that cardiovascular health influences the peripheral and central auditory system [80]. A US retrospective cohort study of 433 adults aged 80 years and over showed that those with cardiovascular morbidity compared to those without had worse hearing (5.47 dB HL higher (worse) mean low-frequency PTA) and experienced a greater rate of decline in low-frequency PTA (by 1.90 dB HL/year vs. 1.18 dB HL/year) [61]. In a population-based cohort study embedded within the Rotterdam Study, the association between carotid atherosclerosis (marker of cardiovascular disease) and hearing loss was investigated in 3724 adults aged ≥45 years [62]. The study factor and outcome were measured using ultrasound and pure-tone audiometry, respectively [62]. More severe atherosclerosis significantly increased the odds of having a higher degree of hearing loss in both the better ear and the right ear [62]. For the better ear, an indicator of this was a higher overall carotid plaque burden (fourth quartile) compared to the lowest (first) quartile of plaque burden (better ear: ordered log-odds (95% CI) = 0.31 (0.08, 0.53); right ear: ordered log-odds (95% CI) = 0.34 (0.11, 0.56). For right-sided hearing loss, overall larger intima-media thickness and larger intima-media thickness in the right carotid increased the odds of a higher degree of hearing loss [ordered log-odds: 0.50 (95% CI: 0.07, 0.92), ordered log-odds: 0.49 (95% CI: 0.10, 0.89), respectively] [62].

## 4. Associations with Other Conditions

Hearing loss is frequently associated with tinnitus, cognitive decline, and dementia [9,10]. For example, tinnitus, which is defined as “the conscious awareness of a tonal or composite noise for which there is no identifiable corresponding external acoustic source” [81], is suggested to be an early symptom of high frequency hearing loss based on associations between tinnitus pitch and edge frequency of the audiogram [10]. Other indicators of association between hearing loss and tinnitus include observed damage to cochlear structures (stria vascularis and outer hair cells), which are implicated in both conditions [10] as well as being strongly associated with increasing age. Evidence from population-based studies has reported that tinnitus prevalence peaks at 11.4 per 10,000 people aged 60–69 years in England [82] and at 14.3% among the same age group in the US [83].

The link between hearing loss, cognitive decline, and dementia is also frequently reported and has gained widespread attention following Livingston et al.’s 2020 report of the Lancet Commission [9]. This report identified hearing loss in mid-life as the biggest modifiable risk factor for a future dementia diagnosis, almost doubling dementia risk (RR = 1.9, 95% CI: 1.4, 2.7) [9]. Earlier studies have also suggested that changes in hearing function may be indicators of age-related cognitive decline 5 to 15 years later [10]. Although further research is needed, it has been hypothesised that hearing loss results in auditory deprivation to the brain, which causes functional and structural dysfunction such as reduced cognitive performance [10] and greater losses in temporal lobe volume compared to individuals with normal hearing [84].

## 5. Future Directions

The evidence presented in this narrative review suggests that lifestyle risk factors for ARHL likely influence oxidative stress pathways to cause hearing loss in adults. Of the studies presented, less than one third reported adjusting for race/ethnicity. As race/ethnicity is a non-modifiable risk factor for ARHL [85] and can significantly influence an individual’s lifestyle choices [86], the variations in the associations between lifestyle risk factors and ARHL reported in this review may be mediated by this factor. However, race/ethnicity were not covariates that were accounted for in the analyses described in most of the studies that we reviewed. Future studies should aim to minimise residual confounding from all relevant variables in statistical analyses to achieve robust findings.

Following on from this narrative review, the recommended next step would be to conduct a systematic review and meta-analysis to quantify the association between oxidative stress and ARHL and the sizes of each lifestyle risk factor. Moreover, as the body of evidence in this area continues to grow, aiding in a better understanding of the potential benefits of modifying lifestyle risk factors, progressing on existing research may provide conclusive outcomes. For instance, animal models have indicated that moderate calorie restriction (around 30%) can lead to improved health and function, as well as protection against age-related diseases [87,88,89,90]. As calorie restriction can transiently increase ROS production, which in turn stimulates a protective response to oxidative stress [87], intervention studies examining the link between moderate calorie restriction and ARHL in humans may provide important data to inform lifestyle recommendations for this condition. Additionally, investigating the impact of intermittent fasting on ARHL may also be worthwhile, as evidence from a systematic review of randomised controlled trials suggests there may be a link to oxidative stress reduction [87]. Moreover, intermittent fasting has gained popularity as a research topic in recent years and is considered a more feasible alternative to calorie restriction as a health promotion strategy [87].

## 6. Conclusions

With ageing demographics, the prevalence of ARHL will increase. Practicing healthy and safe lifestyle behaviours appears to have a role in delaying age-related oxidative damage to the inner ear and, thereby, can help to preserve auditory cells and inner ear function. Although most of the significant associations were small, findings from meta-analyses support a link between ototoxic chemical exposure, obesity, and diabetes with ARHL, while a number of studies also reported a link between noise exposure and smoking with ARHL.

## Figures and Tables

**Figure 1 antioxidants-12-00878-f001:**
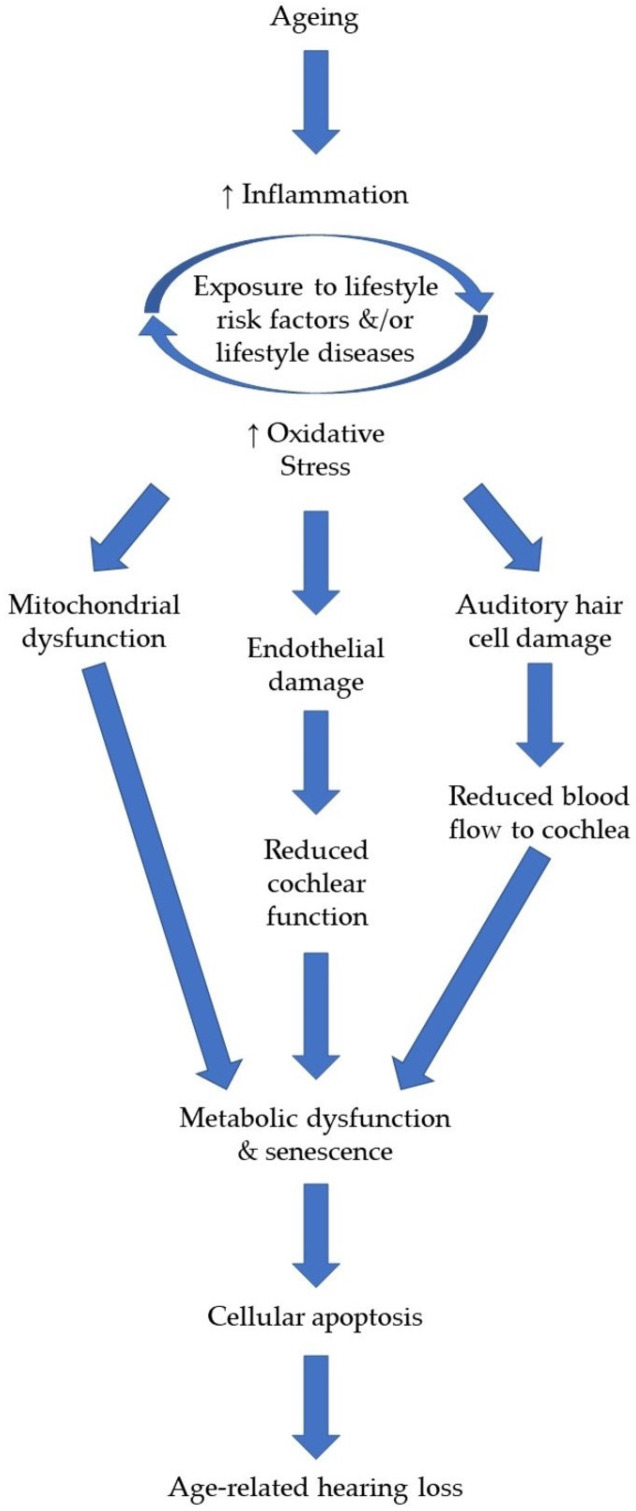
Flowchart of the role of lifestyle risk factors and oxidative stress in ARHL. This figure describes the association between lifestyle risk factors and oxidative stress in ARHL. The proposed mechanism of action relates to dysfunction of the mitochondria, damage to the endothelium and auditory hair cells, leading to apoptosis of auditory cells. ↑ indicates increasing.

**Table 1 antioxidants-12-00878-t001:** Overview of the risk factors and evidence of the association with ARHL.

Risk Factor	Author and Year	Country	Study Design	Key Finding
Noise exposure	Engdahl and Aarhus (2022) [43]	Norway	Repeated cross-sectional study	Exposure to recreational firearms was associated with worse hearing at 3–6 kHz.
	Gopinath et al. (2021) [44]	Australia	Prospective cohort study	Occupational noise exposure for >10 years significantly increased the odds of ARHL by more than two-fold.
	Gates et al. (2000) [45]	United States	Prospective cohort study	Noise-damaged ears showed greater hearing loss decline than non-damaged ears after 15 years.
	Cruickshanks et al. (2010) [46]	United States	Prospective cohort study	No association
	Hederstierna et al. (2016) [47]	Sweden	Prospective cohort study	No association
Ototoxic chemical exposure	Hong et al. (2014) [48]	United States	Cross-sectional study	Lifelong exposure to ototoxic chemicals and/or noise significantly increased the likelihood of hearing loss by two-fold.
	Nakhooda et al. (2019) [49]	South Africa	Systematic review and meta-analysis	Adults in the workplace had a more than two-fold greater risk of hearing loss with combined exposure to ototoxic solvents and noise than to solvents exposure alone.
Smoking	Gopinath et al. (2010) [39]	Australia	Prospective cohort study	Current smokers were 61% more likely to have prevalent hearing loss.
	Cruickshanks et al. (2015) [33]	United States	Prospective cohort study	Smoking significantly increased the risk of incident hearing loss by 31%; former smokers who quit for >5 years had reduced their risk close to a never smoker’s level.
	Lin et al. (2020) [34]	United States	Prospective cohort study	Female smokers had a 21% increased risk of hearing loss; longer duration of smoking cessation was associated with a greater reduction in hearing loss risk.
	Dawes et al. (2014) [32]	United Kingdom	Cross-sectional study	Dose-dependent response observed as longer periods of passive smoke exposure were associated with greater odds of ARHL up to 39% with exposure ≥10 h/week.
Diet quality and eating patterns	Spankovich et al. (2013) [35]	United States	Cross-sectional study	Better diet quality was associated with lower (better) high-frequency hearing thresholds.
Nutrients	Gopinath et al. 2011 [50]	Australia	Prospective cohort study	Higher intakes of vitamin A and E were associated with a 47% reduction in the risk of having moderate or worse hearing loss (>40 dB HL), and a 14% reduction in the prevalence of hearing loss, respectively.
	Péneau et al. (2013) [51]	France	Prospective cohort study	Higher intakes of retinol and vitamin B12 were significantly associated with lower (better) hearing thresholds in a cohort of French women.
	Choi et al. (2014) [52]	United States	Cross-sectional study	Higher intakes of beta-carotene, vitamin C, and magnesium were independently associated with lower (better) hearing thresholds.
	Shargorodsky et al. (2010) [53]	United States	Prospective cohort study	Higher intakes of folate reduced the risk of incident hearing loss by 21% in men ≥60 years.
	Gopinath et al. (2020) [54]	Australia	Prospective cohort study	Higher intakes of isoflavone were associated with a significant reduction in risk of developing ARHL by 36%.
	Dullemeijer et al. (2010) [55]	The Netherlands	Prospective cohort study	High plasma levels of omega-3 fatty acids were significantly associated with less hearing loss at low frequencies.
	Gopinath et al. (2010) [36]	Australia	Prospective cohort study	Higher mean glycaemic index and the lowest cereal fibre intake were 70% more likely to have mild ARHL, higher mean glycaemic load, and carbohydrate intake increased the risk of incident ARHL by 76% and 23%, respectively.
Beverages	Dawes et al. (2014) [32]	United Kingdom	Cross-sectional study	Approximately 40% risk reduction across all levels of alcohol consumption.
	Gopinath et al. (2010) [39]	Australia	Prospective cohort study	Consumption of ≤1 standard alcoholic drink was associated with a 25% risk reduction, and >2 standard drinks reduced the risk by 46%.
	Machado-Fragua et al. (2021) [40]	United Kingdom	Prospective cohort study	Among men, consumption of ≥1 cups of coffee/day lowered their risk of hearing loss by 28%.
Exercise	Curhan et al. (2013) [56]	United States	Prospective cohort study	Higher physical activity intensity and walking ≥2 h/week reduced hearing loss risk by up to 17% and 10%, respectively.
	Kawakami et al. (2022) [42]	Japan	Prospective cohort study	Significant inverse dose-dependent response between increased muscular and performance fitness levels and reduced risk of hearing loss by up to 21%.
	Kawakami et al. (2021) [41]	Japan	Prospective cohort study	Hearing loss risk significantly reduced by 13% with the highest level of exercise intensity (≥525 MET-min/week), and by 11% with 120 min/week physical activity.
Obesity	Yang et al. (2020) [24]	China	Systematic review and meta-analysis	Risk of incident hearing loss increased by 11% with a larger waist circumference, by 15% if overweight, and with each 5 kg/m^2^ increase in BMI, and by 38% if obese.
	Curhan et al. (2013) [56]	United States	Prospective cohort study	There was a dose-dependent association between BMI and self-reported hearing loss, with up to 25% increased risk among adults in obese class III.
	Barrenäs et al. (2005) [57]	Sweden	Prospective register study	Being underweight significantly increased hearing loss risk by 22%.
	Shargorodsky et al. (2010) [58]	United States	Prospective cohort study	Being underweight showed non-significant protective effects for hearing loss.
Diabetes	Horikawa et al. (2013) [59]	Japan	Systematic review and meta-analysis	Overall pooled odds of having any hearing loss were 2.15 times greater among diabetic adults.
	Akinpelu et al. (2014) [60]	Canada	Systematic review and meta-analysis	The odds of incident hearing loss were almost two-fold among type 2 diabetics.
Cardiovascular disease	Wattamwar et al. (2018) [61]	United States	Retrospective cohort study	Presence of cardiovascular morbidity was associated with worse hearing and a greater rate of decline in low-frequency hearing.
	Croll et al. (2019) [62]	The Netherlands	Cross-sectional study	Higher overall carotid plaque burden increased the odds of having a higher degree of hearing loss in the better ear and in the right ear.

## Data Availability

The studies included in this review are available as open access or in peer-reviewed journals.

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
