# Peer review of "A Narrative Review of Lifestyle Risk Factors and the Role of Oxidative Stress in Age-Related Hearing Loss"

_antioxidants, 2023, doi:10.3390/antiox12040878_

Round 1

Reviewer 1 Report

The authors present a broad review of modifiable risk factors associated with ARHL, organized around a model for ARHL that involves inflammation and oxidative damage. I think this is an excellent and timely review and have only minor suggestions for improvement.

1. The model proposed (Figure 1) indicates that ARHL is the result of mitochondrial dysfunction leading to cellular apoptosis. While this mechanism may indeed contribute to ARHL, it is too simplistic to account for ARHL overall. I would like to see the authors expand the model to allow for other contributions, including oxidative stress contributions that are not related directly to mitochondrial dysfunction.

2. Many readers may not be familiar with OR and RR measures or may equate them. Because the review focuses on ORs and RRs, it would be helpful to introduce the abbreviations early on and provide a very brief explanation of what they are and how they differ.

3. L84, This sentence is confusing and it took me a couple of readings to understand it--perhaps you can reword this one to make a clearer link between oxidative stress/inflammation and endothelial health?

4. L136, such as, not such was

5. L239, is the mean difference presented referring to a difference between higher and lower intakes for women or to a difference between women and men?

6. L 433, While hearing devices cannot prevent ARHL, there is evidence to suggest that there may indeed be benefits of amplification for slowing or lessening progressive deterioration (e.g., animal studies using exposure to augmented acoustic environments). Given this, please modify this sentence.

7. L 438, noise exposure more typically involves metabolic, not mechanical, damage to hair cells

8. The review does an excellent job of reporting significant findings in the literature. Most of the significant effects are not large effects, however, so I hope the authors might consider emphasizing (at least in the conclusion section) which factors had a particularly large effect on or association with ARHL.

Author Response

Reviewer 1

Comment 1. The model proposed (Figure 1) indicates that ARHL is the result of mitochondrial dysfunction leading to cellular apoptosis. While this mechanism may indeed contribute to ARHL, it is too simplistic to account for ARHL overall. I would like to see the authors expand the model to allow for other contributions, including oxidative stress contributions that are not related directly to mitochondrial dysfunction.

Response 1. We would like to thank Reviewer 1 for taking the time to provide feedback on this manuscript. We appreciate the suggestion to expand on Figure 1 and incorporate other contributing factors to oxidative stress. This can be found on page 3 of the revised manuscript.

Comment 2. Many readers may not be familiar with OR and RR measures or may equate them. Because the review focuses on ORs and RRs, it would be helpful to introduce the abbreviations early on and provide a very brief explanation of what they are and how they differ.

Response 2. Thank you for your comment. We have taken onboard your suggestion to introduce the risk ratio abbreviations with a brief explanation to help readers understand what they are and mean.

Line 51. “Many of the studies included in this review report on risk ratios. This includes relative risk (RR), which compares the risk of one group relative to another, odds ratio (OR), which indicates the strength of association between one variable relative to another and hazard ratio (HR), which compares the rate of change between one group relative to another 17.”

Comment 3. L84, This sentence is confusing and it took me a couple of readings to understand it--perhaps you can reword this one to make a clearer link between oxidative stress/inflammation and endothelial health?

Response 3. We apologise for the lack of clarity of this sentence and have taken on board your suggestion to reword it to convey a clearer link between oxidative stress, inflammation and endothelial health.

Line 86. “Oxidative stress and inflammation can also cause damage to the endothelium 15. Maintaining the health of the endothelium is important for successful ageing as it is responsible for releasing enzymes and/or nitric oxide to maintain vascular tone, control platelet adhesion, blood clotting, and vascular proliferation 15,21.”

Comment 4. L136, such as, not such was

Response 4. Thank you for identifying this typo. This has been corrected accordingly.

Comment 5. L239, is the mean difference presented referring to a difference between higher and lower intakes for women or to a difference between women and men?

Response 5. We apologise for the lack of clarity of this sentence. The mean difference presented refers to the association between higher intakes of retinol and a significant reduction in decibels HL based on mean difference. We have corrected this in the manuscript as below:

Line 242. “Higher intakes of vitamin A, in the form of retinol, was also found to be significantly associated with lower (better) hearing thresholds in a cohort of French women based on a mean difference of -1.79 decibels hearing level (HL) (95% CI = -3.26, -0.32) compared to the lowest quartile of retinal intake (reference) 51.  The findings in French men were not significant 51. In addition, this study observed improvements in hearing thresholds with higher intakes of vitamin B12 compared to lower intakes (mean difference: -1.57 HL, 95% CI = -3.04, -0.09) 51.”

Comment 6. L 433, While hearing devices cannot prevent ARHL, there is evidence to suggest that there may indeed be benefits of amplification for slowing or lessening progressive deterioration (e.g., animal studies using exposure to augmented acoustic environments). Given this, please modify this sentence.

Response 6. Thank you for your comment. We agree that there is evidence supporting the benefits of hearing device usage to reduce the progression of hearing loss. However, upon consideration of a comment made by Reviewer 2 regarding the refinement of our conclusion to remove mention of new points, we have decided to exclude mention of hearing devices from the conclusion as we have not mentioned these elsewhere in the manuscript.

Comment 7. L 438, noise exposure more typically involves metabolic, not mechanical, damage to hair cells

Response 7. Thank you for your comment. We agree that noise-related hearing damage is typically linked to metabolic activity, and this has been mentioned within the manuscript at Line 122 “This damage triggers significant metabolic activity that can lead to ROS formation, reduce blood flow to the cochlea, and result in cellular apoptosis 16,63.” Upon consideration of Reviewer 2’s comments about reducing the length of our conclusion, and your comment 8 below, we decided to remove L438.

Comment 8. The review does an excellent job of reporting significant findings in the literature. Most of the significant effects are not large effects, however, so I hope the authors might consider emphasizing (at least in the conclusion section) which factors had a particularly large effect on or association with ARHL.

Response 8. Thank you for comment. We have taken on board your suggestion to highlight the more meaningful associations with ARHL in the conclusion as below. We have also created Table 1 which provides an overview of the key associations, so it is easier for the readers to identify the major findings.

Line 468. “With ageing demographics, the prevalence of ARHL will increase. Practicing healthy and safe lifestyle behaviours appear to have a role in delaying age-related oxidative damage to the inner ear and thereby, can help to preserve auditory cells and inner ear function. Although most of the significant associations were small, findings from meta-analyses support a link between ototoxic chemical exposure, obesity and diabetes with ARHL, while a number of studies also consistently reported a link between noise exposure and smoking with ARHL.”

Reviewer 2 Report

Reviewer comments to antioxidants 23-01863

Lifestyle risk factors and the role oxidative stress in age-related hearing loss.

When defining ARHL, in lines 30-32, the authors should be more precise and include the terms bilateral and high frequencies, since ARHL loss initially involves 8, 4 and 2 kHz to extend to all frequencies with the progression of the hair cell damage.

A small section about tinnitus should be included since tinnitus is strongly associated with ARHL and it can be an early symptom of high-frequency hearing loss. This is not mentioned in the manuscript. Please define tinnitus and provide data on the association with ARHL.

Moreover, there is an association of cognitive impairment and ARHL and this also deserves to be mentioned.

The authors are comparing different population-based studies in different countries with divergent outcomes for different lifestyle risk factors. They should consider that 2 major environmental factors are risk factors associated with culture and ethnic background that condition noise exposure and dietary factors. This is relevant for the analysis of nutrients and diseases associated with nutrition such as diabetes, obesity, cardiovascular disease.

Conclusion section must be reduced to 1-2 sentences. This section cannot be an extension of the discussion or future direction section.

The authors should state that they have not used any artificial intelligence tool, such as ChatGPT tool, in the writing or editing of this contribution. 

Suggested references

De Ridder D, Schlee W, Vanneste S et al. Tinnitus and tinnitus disorder: Theoretical and operational definitions (an international multidisciplinary proposal). Prog Brain Res. 2021;260:1-25. doi: 10.1016/bs.pbr.2020.12.002. Epub 2021 Feb 1. PMID: 33637213.

Jafari Z, Kolb BE, Mohajerani MH. Age-related hearing loss and tinnitus, dementia risk, and auditory amplification outcomes. Ageing Res Rev. 2019 Dec;56:100963. doi: 10.1016/j.arr.2019.100963. Epub 2019 Sep 23. PMID: 31557539.

Author Response

Comment 1. When defining ARHL, in lines 30-32, the authors should be more precise and include the terms bilateral and high frequencies, since ARHL loss initially involves 8, 4 and 2 kHz to extend to all frequencies with the progression of the hair cell damage.

Response 1. We would like to thank Reviewer 2 for taking the time to review our manuscript, and appreciate the feedback provided. We agree that the key terms suggested are important when defining ARHL and have revised our definition to reflect this.

Line 31. “It is characterised by bilateral and symmetrical loss of high frequency hearing (≥ 8kHz) in the initial stages, with progression of the condition affecting hearing at lower frequencies due to the irreversible loss of cochlear hair cells and damage to the auditory nerves that are crucial for hearing 11.”

 Comment 2. A small section about tinnitus should be included since tinnitus is strongly associated with ARHL and it can be an early symptom of high-frequency hearing loss. This is not mentioned in the manuscript. Please define tinnitus and provide data on the association with ARHL.

Response 2. Thank you for your comment. We have taken onboard your suggestion to include a small section on the association between hearing loss and tinnitus as below.

Line 421 “For example, tinnitus, which is defined as “the conscious awareness of a tonal or compo-site noise for which there is no identifiable corresponding external acoustic source” 81, is suggested to be an early symptom of high frequency hearing loss based on associations between tinnitus pitch and edge frequency of the audiogram 10. Other indicators of association between hearing loss and tinnitus, include observed damage to cochlear structures (stria vascularis and outer hair cells), which are implicated in both conditions 10 as well as being strongly associated with increasing age. Evidence from population-based studies have reported that tinnitus prevalence peaks at 11.4 per 10,000 people aged 60-69 years in England 82 and at 14.3% among the same age group in the US 83.”

Comment 3. Moreover, there is an association of cognitive impairment and ARHL and this also deserves to be mentioned.

Response 3. Thank you for your comment. We agree the link to cognitive impairment is noteworthy and have mentioned this within the manuscript as below.

Line 430: “The link between hearing loss, cognitive decline and dementia is also frequently reported and has gained widespread attention following Livingston et al.’s 2020 report of the Lancet commission 9. This report identified hearing loss in mid-life as the biggest modifiable risk factor for a future dementia diagnosis, almost doubling dementia risk (RR = 1.9, 95% CI: 1.4, 2.7) 9. Earlier studies have also suggested that changes in hearing function may be indicators of age-related cognitive decline 5 to 15 years later 10. Although further research is needed, it has been hypothesised that hearing loss results in auditory deprivation to the brain which causes functional and structural dysfunction such as reduced cognitive performance 10 and greater losses in temporal lobe volume compared to individuals with normal hearing 84.

Comment 4. The authors are comparing different population-based studies in different countries with divergent outcomes for different lifestyle risk factors. They should consider that 2 major environmental factors are risk factors associated with culture and ethnic background that condition noise exposure and dietary factors. This is relevant for the analysis of nutrients and diseases associated with nutrition such as diabetes, obesity, cardiovascular disease.

Response 4. Thank you for your comment. We agree that this is an important point, as race/ethnicity can influence lifestyle choices and is also a non-modifiable risk factor for ARHL. As we have now identified that less than one third of the studies included in this review accounted for race/ethnicity in their analysis, we have acknowledged that this may explain the variations in associations across the different studies. Therefore, we have also recommended that future studies account for race/ethnicity in their analyses. Unfortunately, as the scope of the review is to focus on modifiable lifestyle risk factors, we are not able to provide more detailed discussions around non-modifiable factors such as ethnicity in this review.

Line 444. “As race/ethnicity is a non-modifiable risk factor for ARHL85 and can significantly influence an individual’s lifestyle choices 86, the variations in the associations between lifestyle risk factors and ARHL reported in this review may be mediated by this factor. However, race/ethnicity was not a covariate that was accounted for in the analysis described in most studies that we reviewed.  Future studies should aim to minimise residual confounding from all relevant variables in statistical analyses to achieve robust findings.”

Comment 5. Conclusion section must be reduced to 1-2 sentences. This section cannot be an extension of the discussion or future direction section.

Response 5. Thank you for your comment. We have taken on board your suggestion to refine our conclusion as below.

Line 468. “With ageing demographics, the prevalence of ARHL will increase. Practicing healthy and safe lifestyle behaviours appear to have a role in delaying age-related oxidative damage to the inner ear and thereby, can help to preserve auditory cells and inner ear function. Although most of the significant associations were small, findings from meta-analyses support a link between ototoxic chemical exposure, obesity and diabetes with ARHL, while a number of studies also consistently reported a link between noise exposure, smoking and ARHL.”

Comment 6. The authors should state that they have not used any artificial intelligence tool, such as ChatGPT tool, in the writing or editing of this contribution.

Response 6. Thank you for your comment. We confirm that ChatGPT was not used in this manuscript, however we have not included this disclaimer within the manuscript as it was not a requirement by the journal. We would be happy to include the statement upon advice from the journal regarding a suitable location to include the text.

Comment 7. Suggested references

  • De Ridder D, Schlee W, Vanneste S et al. Tinnitus and tinnitus disorder: Theoretical and operational definitions (an international multidisciplinary proposal). Prog Brain Res. 2021;260:1-25. doi: 10.1016/bs.pbr.2020.12.002. Epub 2021 Feb 1. PMID: 33637213.
  • Jafari Z, Kolb BE, Mohajerani MH. Age-related hearing loss and tinnitus, dementia risk, and auditory amplification outcomes. Ageing Res Rev. 2019 Dec;56:100963. doi: 10.1016/j.arr.2019.100963. Epub 2019 Sep 23. PMID: 31557539.

Response 7. Thank you for sharing the above references regarding the link between tinnitus, cognition and ARHL. We have included these references while addressing Comments 2 and 3.

Round 2

Reviewer 2 Report

The authors have taken into consideration my comments and suggestions and I have no additional comments. The current version of the manuscript is suitable for publication.